**Subject Category:**
Biology (whole organism)

ecology

carnivore, habituation, livestock,
non-invasive intervention, nuisance animal

**Author for correspondence:**
Igor Khorozyan
e-mail: igor.khorozyan@biologie.
uni-goettingen.de

# How long do anti-predator interventions remain effective? Patterns, thresholds and uncertainty

## Igor Khorozyan and Matthias Waltert

Workgroup on Endangered Species, J.F. Blumenbach Institute of Zoology and Anthropology, Georg-August-Universität Göttingen, Bürgerstrasse 50, Göttingen 37073, Germany

 IK, 0000-0002-0657-7500; MW, 0000-0001-7053-0291

Human–predator conflicts are globally widespread, and effective interventions are essential to protect human assets from predator attacks. As effectiveness also has a temporal dimension, it is of importance to know how long interventions remain most effective and to determine time thresholds at which effectiveness begins to decrease. To address this, we conducted a systematic review of the temporal changes in the effectiveness of non-invasive interventions against terrestrial mammalian predators, defining a temporal trend line of effectiveness for each published case. We found only 26 cases from 14 publications, mainly referring to electric fences ($n = 7$ cases) and deterrents ($n = 7$ cases). We found electric fences and calving control to remain highly effective for the longest time, reducing damage by 100% for periods between three months and 3 years. The effectiveness of acoustical and light deterrents as well as guarding animals eroded quite fast after one to five months. Supplemental feeding was found to be counter-productive by increasing damage over time instead of reducing it. We stress that it is vital to make monitoring a routine requirement for all intervention applications and suggest to standardize periods of time over which monitoring can produce meaningful and affordable information.

## 1. Introduction

Livestock killing by predators, also known as depredation, and nuisance behaviour in human environments often lead to conflicts and retaliatory killing of predators [1–5]. These conflicts are among the main threats to peaceful coexistence of predators and local livelihoods, so that 61% of 28 species of the world's large terrestrial predators already face extinction [6] and many rural and suburban communities still experience strong psychological stress and financial losses to predators [3,7,8]. In regard to prey availability,

human–predator conflicts may occur when natural prey becomes limited and when predator and prey populations recover and more predators need more food [9,10]. In these cases, protection of livestock and human environments from predators requires the application of anti-predator measures or interventions. Non-invasive interventions, which exclude animal handling, should be of high priority to counterbalance social disfavour, high financial burdens and generally low effectiveness of lethal (killing, trapping and poisoning) and invasive non-lethal (translocation, sterilization and shock collars) interventions [11].

In spite of the global importance of anti-predator interventions, very little is known about their effectiveness in reducing damage caused by predators. Respective studies are few, poorly standardized regarding the quantification of damage and intervention effectiveness, often methodologically flawed and biased towards certain predator species [12–16]. Much effort has been made to compile information about evidence-based interventions in relation to wildlife and landscape conservation [17], but those relevant to managing and protecting predators are still rare, not incorporated into global compilations, and scattered across the scientific literature. The small sample size, diversity of species and landscapes addressed, and methodological differences between studies hinder firm conclusions about the effectiveness of interventions against predators [18]. Standardization of anti-predator intervention studies is important to determine the best interventions for a particular target species [16].

The effectiveness of anti-predator interventions tends to decrease over time as predators become habituated to them; therefore, it is of particular importance to know how long interventions remain effective and at which time thresholds the effectiveness begins to decrease. This aspect is studied very insufficiently since few studies monitor effectiveness over time and almost no published studies address and generalize this issue explicitly. The only overview we are aware of is [12], which compared the total duration, but not temporal changes, of the effectiveness of different deterrents. These authors have concluded that chemical, mixed and physical (shock collars) deterrents had the longest lasting effects and acoustical deterrents had the briefest effects. As the effects of deterrents erode quickly due to habituation, it is recommended to use deterrents locally during high-risk short periods like calving or lambing seasons and to apply different interventions one after another to increase the overall effectiveness of applications [12]. Thus, more empirical studies are required to understand the patterns of intervention effectiveness changes over time and to estimate, even preliminarily, the time period when a given intervention remains effective in deterring predators or limiting their access to human assets like livestock and settlements.

Whether the effectiveness of interventions is long lasting or short lasting depends on how fast predators can habituate to them [19,20]. The faster predators habituate to an intervention, the less effective this intervention is. Habituation is a learning process which decreases the animals' responsiveness to a repeated signal and allows them to filter irrelevant information and to adapt [21]. Habituation to anthropogenic factors such as noise, road traffic and visitations allows animals to survive in human-dominated landscapes [22], but it also may reduce the effectiveness of interventions. For example, acoustical and visual deterrents are generally ineffective in human landscapes where noise, light and visual novelties are a norm and animals are quickly adapted to them ([21]; T. Rosen & A. Malkhasyan 2018, personal communication). Unpredictable supply of these stimuli or their modification, such as electrified fladry with visual (flags on ropes) and physical (shock) stimuli combined, tend to decrease habituation and increase the effectiveness of interventions [22,23]. Predator habituation to interventions other than deterrents is seemingly unstudied, but it should become an increasingly common phenomenon as larger and larger tracts of landscapes have been shared by humans and predators and mutual co-adaptation is unavoidable [24]. Individual variation in habituation is also important and interventions can be more effective against shy individuals which are prone to neophobia and avoidance than against bold conspecifics [22,25,26].

It is plausible to surmise that different interventions vary in their ability to cause habituation. For some interventions, habituation can be fast, as in the case of acoustical and visual deterrents mentioned above [12,21], or rather slow if an intervention is effective in the beginning but then predators become habituated to it. The best interventions should cause least habituation over quite a long period of time by supplying strong and long-memorized negatively associated signals, such as electric shock, or by effectively limiting predator access to assets like livestock and garbage dumps. We did not find studies which evaluate anti-predator interventions in this way and suggest that temporal effectiveness changes of interventions deserve more in-depth research.

In this paper, we conduct a systematic review of non-invasive interventions across an array of predator species. We think that this study can be useful for scientists and practitioners involved in human–predator conflicts. For the first time, we (i) develop a framework of interventions causing least habituation, slow habituation and fast habituation as the scenarios of intervention effectiveness change over time; (ii) describe how non-invasive interventions against predators comply with this framework;

**Table 1.** Sample sizes of non-invasive intervention cases used in this study.

| intervention | description | sample size |
|---|---|---|
| electric fences | electric fences encircling the groups of livestock | 7 |
| guarding animals | dogs, llamas and alpacas | 4 |
| calving control | herd management to shorten the calving period | 4 |
| mixed deterrents | pepper spray, rubber bullets and cracker shells with and without dogs, fear-inducing acoustical and visual deterrents | 3 |
| physical deterrents | protective collars and shocking devices | 2 |
| supplemental feeding | supply of carrion | 2 |
| acoustical deterrents | animal sounds | 1 |
| chemical deterrents | lithium chloride (LiCl) | 1 |
| fences | night corrals | 1 |
| herding | presence of shepherd | 1 |

(iii) define non-invasive interventions which are most effective and least effective over time; and (iv) estimate time thresholds at which the effectiveness of non-invasive interventions begins to decrease.

# 2. Material and methods

We collected data on the effectiveness of interventions directed towards the protection of domestic animals, beehives and crops from depredation and local neighbourhoods from nuisance animals. We considered only information related to terrestrial mammalian predators in wild conditions. Collected data are available in electronic supplementary material, Dataset S1.

We used the source literature from the systematic reviews of the effectiveness of predator-targeted interventions [12–16], the online journal *Conservation Evidence* (www.conservationevidence.com, 2004–2018) and the newsletter *Carnivore Damage Prevention News* (www.lcie.org, 2000–2005 and 2014–2016). We also searched for relevant publications in Web of Knowledge (www.webofknowledge.com, 2000–2018) using the keywords 'livestock' AND 'effectiveness' OR 'efficacy' AND *predat*. Additionally, we retrieved relevant papers from Human-Wildlife Conflict Resource Library of the IUCN/SSC Human-Wildlife Conflict Task Force (www.hwctf.org) placed under the key topics 'Electric fences', 'Other barriers', 'Livestock guarding' and 'Deterrents & repellents'.

From the output literature, we selected publications which monitored the effects of interventions and recorded changes in predator-caused damage over time. Changes in damage were reported across the years and months, and for one study [27], we converted two-week damage data into monthly ones. As relevant studies were *a priori* known to be limited, we did not restrict publications to predator species or study durations. The only requirement was to have at least two data points of predator-caused damage with and without interventions in different time periods in order to set a temporal trend of the % of damage reduction. We took the following standardized and most common metrics of damage: number of livestock individuals killed, number of beehive and crop damage records, and number of predator individuals resuming nuisance behaviour after an intervention. We did not use perceived effectiveness, i.e. subjective opinions of farmers about the effectiveness of interventions they apply [28], and used only quantitative metrics of damage.

As the collected information was limited, we considered each study case individually (table 1). Each case described an effect of a particular intervention on protection of a particular asset (livestock, beehives, crops and neighbourhood safety) from a predator species in a study area. Therefore, a source publication could include several cases if they dealt with different interventions, assets, predators or study areas. Each case dealt with one species, two or more species, or an unspecified number of species of livestock and predators as they were described in the source publications. We separated cases for individual sites (farms; [29]) and seasons (spring versus non-spring; [30]) when sufficient information was available.

We quantified the effectiveness of interventions as the % of damage reduction as follows:

$$\% \text{ of damage reduction} = 100 \times (1 - \text{RR}) = 100 \times \left(1 - \frac{A/N_\text{t}}{B/N_\text{c}}\right),$$

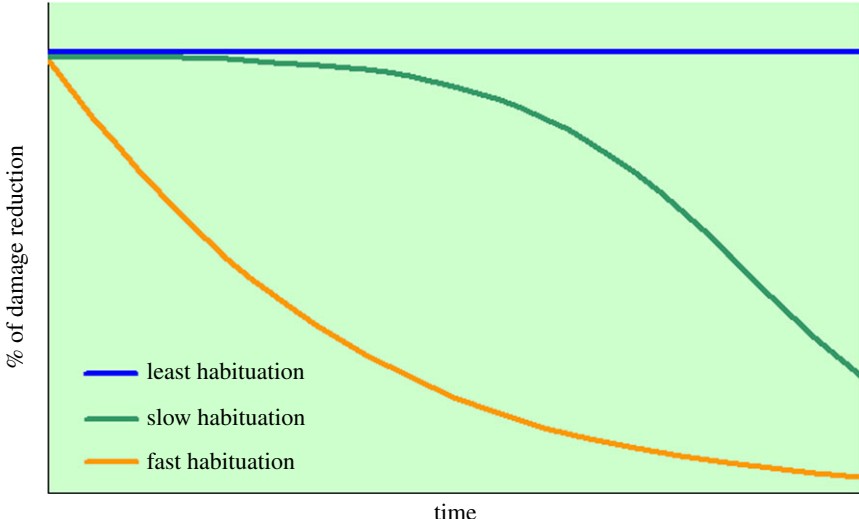

**Figure 1.** The scenarios of effectiveness change over time for interventions causing least habituation, slow habituation and fast habituation by predators.

where RR is the relative risk of damage, $A$ is the metric of damage (e.g. number of livestock individuals killed by predators) with a given intervention, $B$ is the same metric without the intervention, $N_t$ is the treatment sample size (e.g. number of livestock exposed to the intervention) and $N_c$ is the control sample size (e.g. number of livestock not exposed to the intervention or before the intervention is applied) [14]. RR represents a ratio of the probability of damage risk with the intervention to the probability of damage risk without the intervention. Interventions are ineffective at RR > 1, effective at RR < 1 and become most effective at RR = 0 when $A = 0$. In the intervals of damage monitoring where no damage records were obtained in control samples (i.e. $B = 0$), RR was undefined and we excluded these intervals from the analysis [29,30]. For studies using a before–after approach (i.e. the same sample was considered before and after an intervention was applied), we assumed $N_t = N_c$. We used the percentages of $A/N_t$ and $B/N_c$ in calculating the % of damage reduction when they were reported by the authors [31,32]. When the % of damage reduction turned negative, it meant that RR > 1 and that the % of damage increased as a result of a given intervention.

To track the effectiveness trends over time, we calculated the % of damage reduction for each monitoring period (1, 2 years, etc. or one, two months, etc. depending on cases) in an incremental way and studied their trend lines. If control samples covered several monitoring periods [29,30], we incremented the control and treatment samples simultaneously by the same number of steps.

For each study case, we checked how effectiveness trend lines fitted to the scenarios of intervention effectiveness change over time. We suggested these scenarios to depend on how fast predators can adapt and habituate to interventions: (i) interventions causing fast habituation—the % of damage reduction decreases fast as predators become easily habituated and keep on causing damage; (ii) interventions causing slow habituation—the % of damage reduction stays high for some time at the beginning of intervention application, but then predators become habituated and the effectiveness of an intervention goes down; and (iii) interventions causing least habituation—the % of damage reduction is always high or, ideally, maximum (100%) as it is problematic for predators to adapt and get habituated. Interventions causing least habituation may demonstrate an increase in the % of damage reduction if an intervention is imperfect at the beginning, but then its performance improves due to methodological corrections. For example, predators may habituate to electric fences and calving control and kill livestock when these techniques have faults, such as low or no voltage, broken fence or juveniles becoming available, but they do not habituate and stay away when these methods are well-managed [29,33]. Essentially, interventions causing least habituation should limit predator access to livestock and other assets for quite a long period of time. The graphical patterns of these three scenarios are given in figure 1. We checked the trend lines with more than 10 data points for breakpoints using the 'segmented' package v. 0.5-3.0 in R [34]. Each breakpoint represented a time threshold ± s.e. where the effectiveness began to sharply change [34]. We selected the best one-breakpoint or multi-breakpoint models by their Akaike's information criterion (AIC) values, with the lower AIC values indicating better models [35].

# 3. Results

Our search yielded 117 cases from 56 publications, of which only 26 cases from 14 publications contained relevant information and were used in this study (electronic supplementary material, Dataset S1). Twenty cases considered single species and six considered two to three species. Comparatively many cases dealt with the coyote (*Canis latrans*; $n = 9$ cases) and the American black bear (*Ursus americanus*; $n = 5$), and fewer cases with the black-backed jackal (*Canis mesomelas*, $n = 3$), caracal (*Caracal caracal*, $n = 3$), brown bear (*Ursus arctos*, $n = 2$), puma (*Puma concolor*, $n = 2$), domestic dog (*Canis familiaris*, $n = 2$), red fox (*Vulpes vulpes*, $n = 2$), grey wolf (*Canis lupus*, $n = 2$), polar bear (*Ursus maritimus*, $n = 2$), Asiatic black bear (*Ursus thibetanus*, $n = 1$), leopard (*Panthera pardus*, $n = 1$), Iberian lynx (*Lynx pardinus*, $n = 1$) and spotted hyena (*Crocuta crocuta*, $n = 1$). Applied interventions intended to protect cattle ($n = 8$ cases), sheep ($n = 7$) and local neighbourhoods ($n = 5$), as well as livestock in general ($n = 3$), crops and beehives ($n = 1$), ewes ($n = 1$) and lambs ($n = 1$). The most commonly used interventions were deterrents ($n = 7$ cases), electric fences ($n = 7$), calving control ($n = 4$) and guarding animals ($n = 4$) (table 1). The best represented countries were the USA ($n = 11$ cases), Canada ($n = 9$) and South Africa ($n = 3$) and the least represented were Benin, Japan and Spain (one case from each).

The maximum effectiveness of deterrents (100% reduction in damage) was only for periods between three and five months (table 2) and their effectiveness quickly decreased over longer periods (figure 2*a*), thus showing the patterns of interventions causing fast habituation (figure 1). By contrast, calving control reduced damage by 100% during 3 years (table 2). Supplemental feeding by carrion was found to cause fast habituation and remained ineffective, and it even increased damage (figure 2*c*). The effectiveness of supplemental feeding slumped particularly sharply after 13–14 years (breakpoint $13.4 \pm 0.2$ years for spring, case 16 in figure 2*c*; breakpoint $13.6 \pm 0.8$ years for non-spring, case 15 in figure 2*c*).

The duration of the effectiveness of other interventions was variable. The most long-lasting and effective intervention was the electric fence as we recorded five cases with 100% reduction in damage (table 2; cases 6 and 7 in figure 2*b*) during the periods from three months to 3 years. One electric fence application (case 8 in figure 2*b*) showed signs of slow habituation and its effectiveness decreased after 1 year, and another application (case 9 in figure 2*b*) showed a stable but modest 62–65% reduction in damage during 1 year. One case of using night corrals reduced damage by 100% during 2 years (table 2). In most cases, the use of guarding animals showed signs of fast habituation as its effectiveness tended to fall in the next month or year (cases 11, 13 and 14 in figure 2*b*). In case 14, the effectiveness of llamas in protecting lambs from canid depredation decreased after $2.1 \pm 0.3$ months and then fell dramatically after $15.9 \pm 0.5$ months. In one application of llamas protecting ewes from canid depredation (case 10 in figure 2*b*), the effectiveness of guarding animals was maximal during the first five months, then it decreased after $5.0 \pm 0.5$ months, stabilized at $6.6 \pm 0.7$ months and finally slumped after $13.7 \pm 0.9$ months, having demonstrated slow habituation. The only case of herding showed a stable, but low 44–47% reduction in damage during 1 year (case 12 in figure 2*b*).

# 4. Discussion

This study offers a novel framework categorizing non-invasive anti-predator interventions as those causing least habituation, slow habituation and fast habituation by predators (figure 1). Further, it shows how temporal changes in the effectiveness of practically applied interventions fit into this framework. With the consideration of small sample size, which is a reality in intervention effectiveness studies [12–16], and the diversity of predator species and landscapes addressed, we avoid generalizations and attempt to describe the time effectiveness of interventions in a case-specific manner.

We found evidence that electric fences and calving control cause least habituation and remain highly effective for the longest time, namely by reducing damage by 100% during the periods lasting from three months to 3 years. However, this duration of effectiveness could be underestimated and last more than 3 years because the original studies were scheduled to finish earlier than the effectiveness of interventions would begin to erode [12]. The cases of lower effectiveness of electric fences (cases 8 and 9 in figure 2*b*) had insufficient voltage and methodological flaws such as a gate left open [29,33]. High enough voltage is essential to make an electric fence effective and its value depends on predator body size: at least 2.4 kV for coyotes [29], 4.5–5 kV for big cats [44] and up to 200 kV for the largest terrestrial predator, polar bear [33]. The control of calving and lambing seasons is a successful method of reducing depredation as it limits the period of availability of high-risk juveniles to predators to two to three months in comparison with non-seasonal breeding when juveniles are intensively killed throughout a year. Its

**Table 2.** The list of studies which estimated the constant, maximal effectiveness (100% reduction in damage) of non-invasive interventions through regular monitoring. The studies are ranged according to the periods of effectiveness estimation.

| predator | asset | intervention | period of effectiveness estimation | frequency of effectiveness monitoring | source |
|---|---|---|---|---|---|
| puma (Puma concolor) | cattle | calving control | 3 years | annually | [36] |
| grey wolf (Canis lupus) | cattle | calving control | 3 years | annually | [36] |
| American black bear (Ursus americanus) | cattle | calving control | 3 years | annually | [36] |
| coyote (C. latrans) | cattle | calving control | 3 years | annually | [36] |
| coyote (C. latrans) | sheep[a] | electric fence | 3 years | annually | [29] |
| spotted hyena (Crocuta crocuta) | cattle | night corrals | 2 years | annually | [37] |
| Asiatic black bear (U. thibetanus) | crops and beehives | electric fence | 2 years | annually | [38] |
| American black bear (U. americanus) | local neighbourhoods | physical deterrent (shocking device) | 5 months | twice (beginning and end) | [39] |
| grey wolf (C. lupus) | cattle | acoustical and visual deterrents | 3 months | monthly | [40] |
| Iberian lynx (Lynx pardinus) | sheep | electric fence | 3 months | twice (beginning and end) | [41] |

[a]This case indicates the effect of electric fence on sheep depredation on a farm in Canada, and such effects on three other farms in the same study are shown in figure 2.

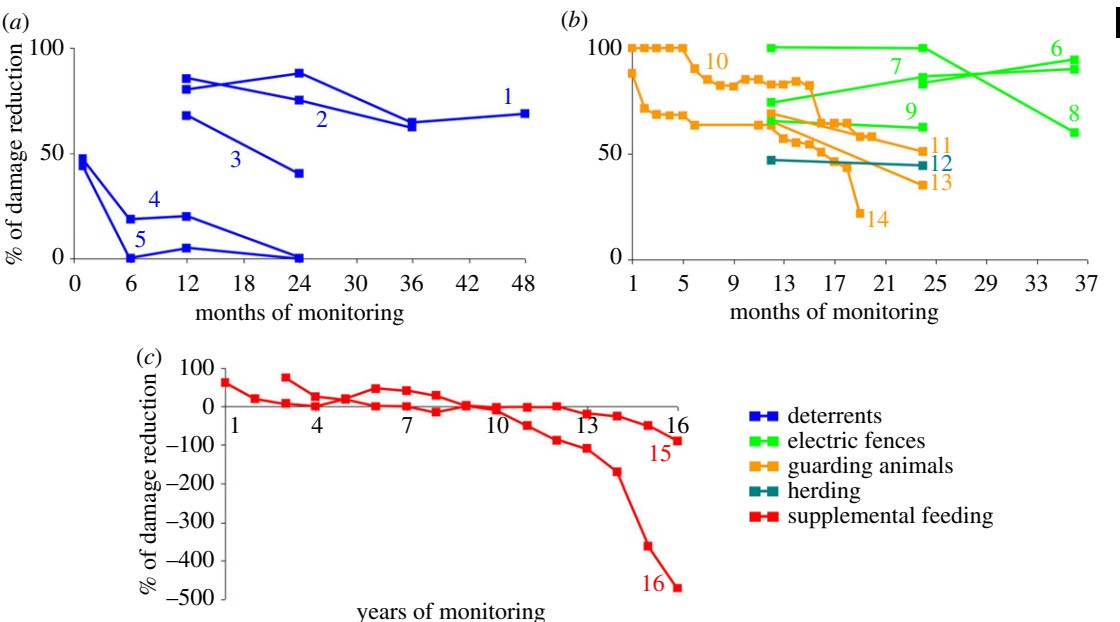

**Figure 2.** The trend lines of the effectiveness of deterrents (*a*), electric fences, guarding animals and herding (*b*), and supplemental feeding (*c*) excluding the cases of constant, maximal 100% damage reduction in table 2. The months and years of effectiveness measurements are marked by squares. Description of cases: 1, effect of acoustical deterrents (aggressive bear sounds) on nuisance behaviour of polar bear (*U. maritimus*) in Canada [33]; 2, effect of a chemical deterrent (LiCl) on sheep depredation by coyote (*C. latrans*) in Canada [31]; 3, effect of a physical deterrent (protective collar) on livestock depredation by black-backed jackal (*C. mesomelas*), caracal (*Caracal caracal*) and leopard (*P. pardus*) in South Africa [32]; 4, effect of a combination of chemical deterrent (pepper spray), physical deterrent (rubber bullet), acoustical deterrent (cracker shell) and dogs on nuisance behaviour of American black bear (*U. americanus*) in the USA [42]; 5, effect of the same deterrents as in case 4, but without dogs, on nuisance behaviour of American black bear in the USA [42]; 6–8, effects of electric fences on sheep depredation by coyote on three farms in Canada [29]; 9, effect of electric fences on nuisance behaviour of polar bear in Canada [33]; 10, effect of guarding llamas (*L. glama*) on ewe depredation by domestic dog (*C. familiaris*), red fox (*V. vulpes*) and coyote in the USA [27]; 11, effect of guarding dogs on livestock depredation by black-backed jackal and caracal in South Africa [32]; 12, effect of herding on sheep depredation by coyote, puma (*Puma concolor*) and American black bear in the USA [43]; 13, effect of guarding alpacas (*Vicugna pacos*) on livestock depredation by black-backed jackal and caracal in South Africa [32]; 14, effect of guarding llamas on lamb depredation by domestic dog, red fox and coyote in the USA [27]; 15, effect of supplemental feeding by carrion on cattle depredation by brown bears (*U. arctos*) during non-spring in Canada [30]; 16, effect of supplemental feeding by carrion on cattle depredation by brown bears during spring in Canada [30].

high effectiveness over time is shown in our study (table 2; [36]) and is also implied from the highest depredation rates during the peak of calving [45,46]. Large-bodied predators are able to kill adult individuals of livestock, but juveniles are still most vulnerable and the control of breeding seasons is essential [44].

We show that acoustical and light deterrents cause fast habituation and they are generally ineffective, as their effectiveness begins to decrease after three to five months (table 2 and figure 2*a*). This is in agreement with previous studies [12], particularly in terms of fast behavioural adaptation of predators to sounds and light in human landscapes [21]. However, some other physical and visual deterrents, such as protective collars and fladry, can be more effective as they target intrinsic ecological habits which are not easy to change. Protective collars fixed on the animal's neck may substantially reduce livestock losses as they provide an effective physical barrier to predators, primarily felids, which kill livestock by throat biting [47]. However, such collars should be ineffective against canids which attack their prey from the hindquarters and flanks. Modest effectiveness of protective collars reported by McManus *et al.* [32] is probably caused by the authors lumping collar effects on depredation by a canid (black-backed jackal) with those by two felids (caracal and leopard) (see case 3 in figure 2*a*). Fladry is a visual deterrent which efficiently limits movements of wolves, but not other predators, in livestock areas, but the information about its effectiveness in reducing livestock losses to depredation is limited [48].

Although guarding animals have been used for millennia and local people widely believe in their effectiveness, we found that the effectiveness of guarding dogs, llamas (*Lama glama*) and alpacas (*Vicugna*

*pacos*) in damage reduction decreased in one to five months and none of these species secured 100% damage reduction even during a short period of time. In the cases describing the application of llamas to protect ewes (case 10 in figure 2*b*) and lambs (case 14 in figure 2*b*) from coyotes, foxes and domestic dogs, a decline of depredation in llama-protected flocks coincided with a decline of depredation in control llama-free flocks which could lead to a deceptive pattern of the decreasing effectiveness of llamas [27]. The effectiveness of dogs is variable depending on their personal traits, training and maintenance [49]. There is a general agreement that llamas outperform dogs in protecting sheep from canids. Llamas show similar predator-deterring behaviour to dogs, but they are harmless to humans, require minimum or no training, establish strong bonds with small stock, do not need special maintenance conditions, live longer than dogs and their keeping is economical as it is best to have only one llama (large gelded male) per flock [27,50,51]. However, it remains unknown whether llamas can successfully protect cattle which disperse much wider than small stock while grazing, and whether they can effectively deter predators other than canids. Alpacas and llamas are intrinsically aggressive towards canids; therefore, it is possible that the mediocre 35–67% damage reduction by alpacas (case 13 in figure 2*b*) could be caused by lumping depredation by black-backed jackals with that by caracals [32]. Also, being smaller than llamas, alpacas can successfully protect lambs, but possibly less so adult sheep [52].

Supplemental feeding by carrion was planned originally to reduce livestock losses by brown bears as these predators were expected to consume high-quality and easily available food and thus have less incentives to seek and attack livestock. In practice, the result was opposite as bears even increased the numbers of livestock killed over years, especially after 13–14 years, because feeding sites were visited by only few bears and the bear population tended to increase and spread regardless of supplemental feeding [30]. A similar situation was observed in a study [53] where carrion from feeding sites did not play a major role in bear diet and depredation kept on increasing along with bear population and livestock numbers.

Herding and construction of night corrals have been used traditionally to protect livestock from predators, but surprisingly little is known about the effectiveness of these interventions and even less about how this effectiveness changes over time. In our study, we had only one case with 100% damage reduction by corrals during 2 years (table 2; [37]) and one case of modest 44–47% damage reduction by herders (case 12 in figure 2*b*; [43]). We do not know other studies which measure the corral effectiveness, with or without temporal trends. In [43], the effectiveness of herders was quite low and further declined because they were busy with other duties like fence maintenance and could not spend more time with livestock. Apart from the lack of attentiveness, other factors that may make herding ineffective and even counterproductive are the failure to deter predators [54], low number of shepherds per herd [55] and herding by children [56].

This study has several limitations. First, there is still a paucity of scientific literature on the effectiveness of anti-predator interventions [12–16,18]. The effectiveness has been monitored very seldom, and in most studies it is measured only once in the end of the study, which makes it impossible to find how the effectiveness changes over time during the study. For this reason, we considered each case individually and did not apply more sophisticated methods of data analysis like modelling. Also, as there were only one to nine cases per predator species in our study, we did not analyse the effectiveness of interventions across the predator species. Second, the monitoring of intervention effectiveness has been done inconsistently through different time intervals, from two weeks and one month to 1 year (figure 2). There is a vital need to make the monitoring a routine requirement for all intervention applications and to standardize the periods of time over which the monitoring can produce meaningful and affordable information. Measuring the effectiveness once every quarter or half a year seems to be most practical as it allows to collect sufficient damage records with reasonable investments and to account for the effects of seasonality in livestock management (e.g. transhumance) and predator ecology (e.g. cub rearing) on the effectiveness of interventions. Third, there is a possibility that this study could be affected by publication bias which makes positive results more likely to be published and thus can overestimate the effectiveness of interventions. Although publication bias is always possible [57], we believe that its effect on our study was minimized by the balanced inclusion of studies with high, low and no effectiveness.

# 5. Conclusion

Temporal changes in the effectiveness of interventions striving to protect humans and their assets, such as livestock, from predators are studied very insufficiently and our systematic review has revealed only 26

cases from 14 publications. Due to small sample size and the diversity of predators and landscapes addressed, we avoided generalizations but were able to associate the effectiveness of each studied case with least, slow and fast habituation by predators. We found that electric fences and calving control caused least habituation and remained highly effective during several years. By contrast, the effectiveness of deterrents and guarding animals eroded over several months and supplemental feeding increased livestock depredation instead of reducing it; therefore, these interventions caused slow to fast habituation by predators. More studies are required to study temporal changes of the effectiveness of such popular interventions as herding and night corrals. From a methodological standpoint, application of interventions requires careful and regular monitoring at standardized time intervals.

Data accessibility. The original Dataset S1 collected for this study is available within the Dryad Digital Repository at: https://doi.org/10.5061/dryad.p6k2cb0 [58].

Authors' contributions. I.K. conceived the study design, collected and analysed data, and drafted the manuscript. M.W. coordinated the study, participated in the design of the study and helped draft the manuscript. All authors gave final approval for publication.

Competing interests. The authors declare no competing interests.

Funding. This study was supported by German Research Foundation (Deutsche Forschungsgemeinschaft, DFG, grant no. WA 2153/5-1).

Acknowledgements. We thank M. Filla for assistance with segmented regression analysis in R and two anonymous reviewers for thoughtful comments, which greatly improved the quality of the paper.

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
