## [Reviewer comments · Royal Society Open Science]

Review History

RSOS-190826.R0 (Original submission)

Review form: Reviewer 1

Is the manuscript scientifically sound in its present form?

Yes

Are the interpretations and conclusions justified by the results?

Yes

Is the language acceptable?

Yes

Is it clear how to access all supporting data?

Yes

Do you have any ethical concerns with this paper?

No

Have you any concerns about statistical analyses in this paper?

No

Recommendation?

Accept with minor revision (please list in comments)

Comments to the Author(s)

General comments

In this paper, the authors conduct a systematic review to assess how the effectiveness of interventions to deter predators from interfering in human environments changes over time. The amount of data the authors found seems to have been quite limited. However, despite the limited data, I think they made a convincing case that interventions that predators readily habituate to (such as light and acoustic deterrents, or guard animals) become less effective over time at a much more rapid rate than interventions that physically prevent predators from encountering resources (such as electric fencing and calving control). They also highlight a need for more research on the long term effectiveness of anti-predator interventions.

One thing that stood out to me was that the authors suggest that the interventions that remained effective did so because the predators could not habituate to them (Introduction line 22 pg. 3, Figure 1). However, it is not entirely clear to me that a lack of habituation by the predator explains why some of the interventions the authors discuss remained effective over time. For example, a predator could habituate to the presence of an electric fence in its environment, but it still wouldn't be able to get through the fence to kill livestock. In the case of calving control, learning by the predator may even enhance the effectiveness of the intervention – As the predators learn that easy-to-eat juvenile individuals are not usually present at a livestock holding area, they stop looking at the livestock holding area for food.

One thing I think the authors could do to improve the paper is to provide more details on the paper selection process. Right now, the authors provide details about the search procedure and the places they searched for papers (Methods, lines 15-24 pg. 4). However the details of the study selection protocol are limited to a single statement “From the output literature, we selected publications which monitored the effects of interventions and recorded changes in predator-caused damage over time” (Methods, lines 25-26 pg. 4). I think a more complete description of the criteria by which studies were included or eliminated from the review would be helpful. Were the included publications limited to certain predators only? Was there any assessment of study quality when deciding which studies to include? Was there a minimum length of monitoring that a study needed to be included, and was there a minimum number of assessments of intervention effectiveness a study had to contain to be included?

The sample of studies included was small, which the authors explain by indicating that there is limited research available on the changes in effectiveness of anti-predator interventions over time. In addition to adding details on study selection, I think the authors may want to strongly consider adding in information about the number of studies returned by their initial searches, and the number of studies that could not be used in the review due to failure to meet the criteria for inclusion. Adding in this information would add some numerical support for the assertion that there is insufficient research on the long-term effectiveness of anti-predator interventions and how animals habituate to them. It may also give readers a better sense of what is currently in the literature, and what needs to be added.

Finally, I thought the classification of interventions into the “aversion”, “husbandry”, and “management” categories was somewhat arbitrary. For example, the shocking device described by Breck et. al (2006) to protect food sources from black bears (# 8 in the supplementary materials) is in the “aversion” category, while an electric fence used to protect cattle from

predators is considered a “husbandry” intervention. The discussion sections makes no mention of these categories, and I think the authors could simply eliminate them without any harm to the paper or the points they make about the effectiveness of different interventions.

Specific comments

Summary:

- Line 3, pg 1 – the “for” and “do” in the second sentence on the line could be removed, and “know replaced with “learn” resulting in a clearer “To learn how long interventions remain effective, we conducted a systematic review, estimated the effectiveness of noninvasive interventions against terrestrial mammalian predators and compared temporal changes of this effectiveness”

Introduction:

- Line 44, pg 1 – How is a “large animal” predator defined?
 - Line 46, pg 1 – “These conflicts are among the main threats to peaceful co-existence of conservation and local livelihoods” – Did you mean coexistence of predators and local livelihoods?

Tables and figures:

- Table 2:

-- I don’t think the notation “farm B” will mean anything to readers unless they look at the supplementary materials or the original publication
 -- The time scales of the cases in the table are quite different, and it was not clear to me what criteria were used to assess whether an intervention achieved 100% effectiveness. For example, the acoustical and visual deterrents against grey wolves (case 9 supplementary materials) were included with a period of effectiveness of 3 months, with no measurements made after three months. However, the guard llamas used to protect sheep (case 20 supplementary materials) were not included despite maintaining 100% effectiveness for 5 months, presumably because the effectiveness declined sharply from months 6-20. Why wasn’t this case included in the table with a period of effectiveness estimate of 5 months?
 - Figure 2 – Given the small sample size, I fully understand why the authors present each case separately. However, I think it would be easier to follow along with the discussion of the evidence of the efficacy of each intervention type if the authors considered using one color or symbol for each of the intervention types, (i.e Green for electric fence, purple for physical deterrence ect.).

Review form: Reviewer 2

Is the manuscript scientifically sound in its present form?

Yes

Are the interpretations and conclusions justified by the results?

Yes

Is the language acceptable?

No

Is it clear how to access all supporting data?

No

Do you have any ethical concerns with this paper?

No

Have you any concerns about statistical analyses in this paper?

I do not feel qualified to assess the statistics

Recommendation?

Major revision is needed (please make suggestions in comments)

Comments to the Author(s)

Please find the attached document for my comments (Appendix A).

Decision letter (RSOS-190826.R0)

22-Jul-2019

Dear Dr Khorozyan,

The editors assigned to your paper ("How long do anti-predator interventions remain effective? Patterns, thresholds and uncertainty") have now received comments from reviewers. We would like you to revise your paper in accordance with the referee and Associate Editor suggestions which can be found below (not including confidential reports to the Editor). Please note this decision does not guarantee eventual acceptance.

Please submit a copy of your revised paper before 14-Aug-2019. Please note that the revision deadline will expire at 00.00am on this date. If we do not hear from you within this time then it will be assumed that the paper has been withdrawn. In exceptional circumstances, extensions may be possible if agreed with the Editorial Office in advance. We do not allow multiple rounds of revision so we urge you to make every effort to fully address all of the comments at this stage. If deemed necessary by the Editors, your manuscript will be sent back to one or more of the original reviewers for assessment. If the original reviewers are not available, we may invite new reviewers.

If your study uses humans or animals please include details of the ethical approval received, including the name of the committee that granted approval. For human studies please also detail

whether informed consent was obtained. For field studies on animals please include details of all permissions, licences and/or approvals granted to carry out the fieldwork.

- Data accessibility

If you wish to submit your supporting data or code to Dryad (<http://datadryad.org/>), or modify your current submission to dryad, please use the following link:
<http://datadryad.org/submit?journalID=RSOS&manu=RSOS-190826>

- Competing interests

- Authors' contributions

- Acknowledgements

- Funding statement

Kind regards,
Andrew Dunn
Royal Society Open Science Editorial Office

on behalf of Dr Punidan Jeyasingh (Associate Editor) and Kevin Padian (Subject Editor)
openscience@royalsociety.org

Associate Editor's comments (Dr Punidan Jeyasingh):

Associate Editor: 1

Comments to the Author:

This manuscript analyzes the effectiveness of methods to deter wild predators from affecting human assets. It is a timely topic, of much importance. The manuscript was reviewed by two experts, both of whom were enthusiastic about the topic and the manuscript in general. Nevertheless, both reviewers raised several issues that need to be carefully addressed. I felt the reviews were fair and constructive. With much gratitude to the reviewers, I invite the authors to make these revisions and resubmit a new version for further evaluation.

Comments to Author:

Reviewers' Comments to Author:

Reviewer: 1

Comments to the Author(s)

General comments

In this paper, the authors conduct a systematic review to assess how the effectiveness of interventions to deter predators from interfering in human environments changes over time. The amount of data the authors found seems to have been quite limited. However, despite the limited data, I think they made a convincing case that interventions that predators readily habituate to (such as light and acoustic deterrents, or guard animals) become less effective over time at a much more rapid rate than interventions that physically prevent predators from encountering resources (such as electric fencing and calving control). They also highlight a need for more research on the long term effectiveness of anti-predator interventions.

One thing that stood out to me was that the authors suggest that the interventions that remained effective did so because the predators could not habituate to them (Introduction line 22 pg. 3, Figure 1). However, it is not entirely clear to me that a lack of habituation by the predator explains why some of the interventions the authors discuss remained effective over time. For example, a predator could habituate to the presence of an electric fence in its environment, but it still wouldn't be able to get through the fence to kill livestock. In the case of calving control, learning by the predator may even enhance the effectiveness of the intervention - As the predators learn that easy-to-eat juvenile individuals are not usually present at a livestock holding area, they stop looking at the livestock holding area for food.

One thing I think the authors could do to improve the paper is to provide more details on the paper selection process. Right now, the authors provide details about the search procedure and the places they searched for papers (Methods, lines 15-24 pg. 4). However the details of the study selection protocol are limited to a single statement "From the output literature, we selected publications which monitored the effects of interventions and recorded changes in predator-caused damage over time" (Methods, lines 25-26 pg. 4). I think a more complete description of the criteria by which studies were included or eliminated from the review would be helpful. Were the included publications limited to certain predators only? Was there any assessment of study quality when deciding which studies to include? Was there a minimum length of monitoring that a study needed to be included, and was there a minimum number of assessments of intervention effectiveness a study had to contain to be included?

The sample of studies included was small, which the authors explain by indicating that there is limited research available on the changes in effectiveness of anti-predator interventions over time. In addition to adding details on study selection, I think the authors may want to strongly consider adding in information about the number of studies returned by their initial searches, and the number of studies that could not be used in the review due to failure to meet the criteria for inclusion. Adding in this information would add some numerical support for the assertion that there is insufficient research on the long-term effectiveness of anti-predator interventions and how animals habituate to them. It may also give readers a better sense of what is currently in the literature, and what needs to be added.

Finally, I thought the classification of interventions into the “aversion”, “husbandry”, and “management” categories was somewhat arbitrary. For example, the shocking device described by Breck et. al (2006) to protect food sources from black bears (# 8 in the supplementary materials) is in the “aversion” category, while an electric fence used to protect cattle from predators is considered a “husbandry” intervention. The discussion sections makes no mention of these categories, and I think the authors could simply eliminate them without any harm to the paper or the points they make about the effectiveness of different interventions.

Specific comments

Summary:

- Line 3, pg 1 – the “for” and “do” in the second sentence on the line could be removed, and “know replaced with “learn” resulting in a clearer “To learn how long interventions remain effective, we conducted a systematic review, estimated the effectiveness of noninvasive interventions against terrestrial mammalian predators and compared temporal changes of this effectiveness”

Introduction:

- Line 44, pg 1 – How is a “large animal” predator defined?
- Line 46, pg 1 – “These conflicts are among the main threats to peaceful co-existence of conservation and local livelihoods” – Did you mean coexistence of predators and local livelihoods?

Tables and figures:

- Table 2:

- I don’t think the notation “farm B” will mean anything to readers unless they look at the supplementary materials or the original publication
- The time scales of the cases in the table are quite different, and it was not clear to me what criteria were used to assess whether an intervention achieved 100% effectiveness. For example, the acoustical and visual deterrents against grey wolves (case 9 supplementary materials) were included with a period of effectiveness of 3 months, with no measurements made after three months. However, the guard llamas used to protect sheep (case 20 supplementary materials) were not included despite maintaining 100% effectiveness for 5 months, presumably because the effectiveness declined sharply from months 6-20. Why wasn’t this case included in the table with a period of effectiveness estimate of 5 months?
- Figure 2 – Given the small sample size, I fully understand why the authors present each case separately. However, I think it would be easier to follow along with the discussion of the evidence of the efficacy of each intervention type if the authors considered using one color or symbol for each of the intervention types, (i.e Green for electric fence, purple for physical deterrence ect.).

Reviewer: 2

Comments to the Author(s)

Please find the attached document for my comments.

Author's Response to Decision Letter for (RSOS-190826.R0)

See Appendix B.

Decision letter (RSOS-190826.R1)

20-Aug-2019

Dear Dr Khorozyan,

I am pleased to inform you that your manuscript entitled "How long do anti-predator interventions remain effective? Patterns, thresholds and uncertainty" is now accepted for publication in Royal Society Open Science.

on behalf of Dr Punidan Jeyasingh (Associate Editor) and Kevin Padian (Subject Editor)
openscience@royalsociety.org

Associate Editor Comments to Author (Dr Punidan Jeyasingh):

I thank the authors for carefully addressing reviewer comments. The manuscript is much improved, and I'm happy to recommend publication.

Appendix A

Review for “how long do anti-predator interventions remain effective? Patterns, thresholds and uncertainty.”

General comments

The paper addresses a topical subject and has relevance. It seeks novel ways to determine how long deterrents remain effective.

Generally, I found this paper to make statements without really going into enough background literature at the international level to support the claims it does. For example, Page 2, line 1 “ Despite of the huge effort in compiling evidence-based conservation interventions, those relevant for managing and protecting predators are still not incorporated [13]. This adds uncertainty and makes results of reviews of anti-predator interventions inconclusive [14].” This is very general to state that no literature incorporates managing and protecting predators are not incorporated is not true, and that reviews of anti-predator interventions are inconclusive is misleading. I suggest the latter is better broken down into why these may be inconsistent as there are a range of issues in this regard. This is important as the topic is complex dealing with many species across varying environmental conditions.

I think that more studies are available to incorporate to this study to increase the value of this manuscript.

The new methods are interesting and the authors should provide the code and models used to assist future work and make this comparative for future work.

Overall, I found that the paper needs overall improvements in terms of 1) grammar throughout the manuscript, and 2) I found it a bit concerning that more studies that appear to suit the criteria to this review, were not included. Additionally, I think that the findings have value, however, the methods require more detail and providing the models will help readers understand this better.

Abstract

It seems that the authors may not use English as a first language, and in its current form, the manuscript will require major grammatical editing throughout, by an editor with better experience with the language.

For example, page 2 lines 14-15, “To know for how long do interventions remain effective...” would better read “To know how long interventions remain effective for..”. Then again in lines 21 and 22, “We found electric fences...100% during up to three years” indicates little effort

on editing this document. This continues throughout the manuscript and should be improved prior to publication.

Page 1, line 22 – 23, “The effectiveness ...eroded quite fast after 1 – 5 months...” I would like to see the range or average here if that is possible.

Line 25 – 26, In which way was night corrals and herding inclusive? Was it inclusive of short term effectiveness? This is an odd finding compared to other literature. What would be the reasons for this in this case?

Introduction

Line 33, I would suggest more international studies as references to this broad problem statement.

Line 39 – 41, “Human-predator conflict may occur...”. Firstly, the grammar is very poor. Also, these are certainly not the ONLY two instances carnivore-wildlife conflict occur. It is misleading in its current statement. Perhaps the authors are introducing some examples of when conflict occurs? This needs to be stated as examples, rather than the ONLY two scenario’s for human-carnivore conflict situations.

Page 2, line one please see general comments section above.

Page 2, lines 7 – 10, suggests that there is no evidence in literature discussing duration of effect. There are more studies which assess and discuss depredation over time using various methods albeit for certain periods, but these have relevance to this manuscript. See below a few papers I found on the duration effect and I’m sure there will be more I didn’t have the time to look for more:

- **Tools for the Edge: What's New for Conserving Carnivores**
John A. Shivik *BioScience*, Volume 56, Issue 3, March 2006, Pages 253–259”. Looked at several tools with various findings of duration of effect.
- Non-lethal defence of livestock against predators: flashing lights deter puma attacks in Chile, Ohrens et al., 2019, found that fox lights worked for at least 4 months.
- Davidson-Nelson SJ and Gehring TM. 2010. Testing fladry as a nonlethal management tool for wolves and coyotes in Michigan. *Human-Wildlife Interactions* 4: 87–94.
- Gehring TM, VerCauteren KC, and Cellar AC. 2010. Good fences make good neighbors: implementation of electric fencing for establishing effective livestock protection dogs. *HumanWildlife Interactions* 4: 144–49. Gehring TM, VerCauteren KC, Provost ML, et al. 2010. Utility of livestock-protection dogs for deterring wildlife from cattle farms. *Wildlife Res* 37: 715–21.

- Espuno N, Lequette B, Poulle ML, et al. 2004. Heterogeneous response to preventive sheep husbandry during wolf recolonization of the French Alps. *Wildlife Society B* 32: 1195–208.

The point above along with the couple examples I found relatively quickly, has relevance to page 2, lines 17 – 21.

Lines 22 – page 3 lines 6 lead the reader more clearly into what is being assessed. I suggest the authors bring this in earlier in the introduction and summarise this better in the abstract, as I had difficulty in understanding the objectives until this point.

Methods

Page 3, line 18, why was the method limited to two search periods i.e. 2000-2005 and then from 2014-2016? Why was there a period gap and why did this search not look until 2018 like the other sites?

An important problem with ‘effectiveness’ studies is that many studies are not based on verified losses, rather opinion. It would be of value to compare these two groups of studies differently as many don’t consider opinion surveys realistic to the ‘effectiveness’ question due to bias or placebo effect for example.

The algorithm designed for this is new and appears to be relatively transferable to other studies. However:

- 1) How was Nc data available in each of these cases? It is rarely listed in deterrent studies.
- 2) Building such models adds vagueness to the study results. I suggest that the models and code developed be added as a supplementary 2 document to assist future work. This may also assist researches to identify variables which cause more or less variation in results.

Discussion

Consider placing paragraph two lines 24 – 29 in results section.

Line 53, was it an expectation to find guarding animals more effective? Why?

Line 55. Did you want to see 100% reduction in methods? That is a reach for any method, however, rather it may be useful for the reader to know if there was a significant decline, or a difference in decline in depredation over that period of time as opposed to total decline?

Increasing the studies may help this manuscript. Studies do seem to be available relating to duration and livestock decrease over time in the list mentioned below are several, but more

are available that are not included in this study. I understand it is not easy finding these, however, the small sample size does limit impact this paper can make. Also, it is important to find or separate studies which consider actual losses versus perceived losses.

Appendix B

Associate Editor: 1

Comments to the Author:

This manuscript analyzes the effectiveness of methods to deter wild predators from affecting human assets. It is a timely topic, of much importance. The manuscript was reviewed by two experts, both of whom were enthusiastic about the topic and the manuscript in general. Nevertheless, both reviewers raised several issues that need to be carefully addressed. I felt the reviews were fair and constructive. With much gratitude to the reviewers, I invite the authors to make these revisions and resubmit a new version for further evaluation.

We thank the editor and both reviewers for the positive feedback. We have thoroughly revised the paper according to all comments, responded to each comment, and made changes in the text which are highlighted in yellow. Our responses are shown in red.

Reviewer: 1

Comments to the Author(s)

General comments

In this paper, the authors conduct a systematic review to assess how the effectiveness of interventions to deter predators from interfering in human environments changes over time. The amount of data the authors found seems to have been quite limited. However, despite the limited data, I think they made a convincing case that interventions that predators readily habituate to (such as light and acoustic deterrents, or guard animals) become less effective over time at a much more rapid rate than interventions that physically prevent predators from encountering resources (such as electric fencing and calving control). They also highlight a need for more research on the long term effectiveness of anti-predator interventions.

One thing that stood out to me was that the authors suggest that the interventions that remained effective did so because the predators could not habituate to them (Introduction line 22 pg. 3, Figure 1). However, it is not entirely clear to me that a lack of habituation by the predator explains why some of the interventions the authors discuss remained effective over time. For example, a predator could habituate to the presence of an electric fence in its environment, but it still wouldn't be able to get through the fence to kill livestock. In the case of calving control, learning by the predator may even enhance the effectiveness of the intervention – As the predators learn that easy-to-eat juvenile individuals are not usually present at a livestock holding area, they stop looking at the livestock holding area for food.

The studies show that predators generally do not habituate to electric fences and calving control, but continue to warily move in these areas and resume livestock killing once favorable conditions emerge, such as when voltage is off or low, fences have breakage, or juveniles become available. We believe that predators become habituated to these techniques only when they are poorly managed. We added new sentences in Methods about this.

One thing I think the authors could do to improve the paper is to provide more details on the paper selection process. Right now, the authors provide details about the search procedure and the places they searched for papers (Methods, lines 15-24 pg. 4). However the details of the study selection protocol are limited to a single statement “From the output literature, we selected publications which monitored the effects of interventions and recorded changes in predator-caused damage over time” (Methods, lines 25-26 pg. 4). I think a more complete description of

the criteria by which studies were included or eliminated from the review would be helpful. Were the included publications limited to certain predators only? Was there any assessment of study quality when deciding which studies to include? Was there a minimum length of monitoring that a study needed to be included, and was there a minimum number of assessments of intervention effectiveness a study had to contain to be included?

As relevant studies were a priori known to be limited, we did not restrict publications to predator species or study durations. The only requirement was to have at least two data points of predator-caused damage with and without interventions in different time periods in order to set a temporal trend of the % of damage reduction. We have added this sentence in the text.

The sample of studies included was small, which the authors explain by indicating that there is limited research available on the changes in effectiveness of anti-predator interventions over time. In addition to adding details on study selection, I think the authors may want to strongly consider adding in information about the number of studies returned by their initial searches, and the number of studies that could not be used in the review due to failure to meet the criteria for inclusion. Adding in this information would add some numerical support for the assertion that there is insufficient research on the long-term effectiveness of anti-predator interventions and how animals habituate to them. It may also give readers a better sense of what is currently in the literature, and what needs to be added.

Our search yielded 117 cases from 56 publications, of which only 26 cases from 14 publications contained relevant information and were used in this study. We added this sentence in the beginning of Results.

Finally, I thought the classification of interventions into the “aversion”, “husbandry”, and “management” categories was somewhat arbitrary. For example, the shocking device described by Breck et. al (2006) to protect food sources from black bears (# 8 in the supplementary materials) is in the “aversion” category, while an electric fence used to protect cattle from predators is considered a “husbandry” intervention. The discussion sections makes no mention of these categories, and I think the authors could simply eliminate them without any harm to the paper or the points they make about the effectiveness of different interventions.

We have removed these categories throughout the paper and its original Dataset S1 file. A new version of the dataset is uploaded on Dryad.

Specific comments

Summary:

- Line 3, pg 1 – the “for” and “do” in the second sentence on the line could be removed, and “know replaced with “learn” resulting in a clearer “To learn how long interventions remain effective, we conducted a systematic review, estimated the effectiveness of noninvasive interventions against terrestrial mammalian predators and compared temporal changes of this effectiveness”

Done.

Introduction:

- Line 44, pg 1 – How is a “large animal” predator defined?

We deleted the phrase about large animals.

- Line 46, pg 1 – “These conflicts are among the main threats to peaceful co-existence of conservation and local livelihoods” – Did you mean coexistence of predators and local livelihoods?

We changed “conservation” to “predators”. Originally, we implied that depredation and nuisance behavior of predators also lead to the conflict between conservation and human livelihoods because conservation is focused on protecting predators and other wildlife, with less attention being paid to livelihoods and socio-economic interests of local people.

Tables and figures:

- Table 2:

-- I don't think the notation “farm B” will mean anything to readers unless they look at the supplementary materials or the original publication

The case of farm B is indicated in Table 2 and the other three cases (three farms) from the same study are shown in Fig. 2. We deleted “farm B” in Table 2, added the asterisk and placed the note in table legend.

-- The time scales of the cases in the table are quite different, and it was not clear to me what criteria were used to assess whether an intervention achieved 100% effectiveness. For example, the acoustical and visual deterrents against grey wolves (case 9 supplementary materials) were included with a period of effectiveness of 3 months, with no measurements made after three months. However, the guard llamas used to protect sheep (case 20 supplementary materials) were not included despite maintaining 100% effectiveness for 5 months, presumably because the effectiveness declined sharply from months 6-20. Why wasn't this case included in the table with a period of effectiveness estimate of 5 months?

We depicted the cases of effectiveness estimated for the whole study and did not break up the same study into the period of 100% effectiveness and the subsequent period of effectiveness changes. We did this to avoid confusion and misunderstanding.

Table 2 indicates only the cases which retained 100% effectiveness throughout the study, therefore the gray wolf case from Breck et al. (2002) was included in Table 2. The llama case from Meadows and Knowlton (2000), case 20 in Dataset S1, started with 100% effectiveness but then the effectiveness went down to 58.2%. As the effectiveness in this case was not 100% all the time, we placed it in Fig. 2 as case 10.

- Figure 2 – Given the small sample size, I fully understand why the authors present each case separately. However, I think it would be easier to follow along with the discussion of the evidence of the efficacy of each intervention type if the authors considered using one color or symbol for each of the intervention types, (i.e Green for electric fence, purple for physical deterrence ect.).

We changed the colors of the intervention effectiveness lines as suggested. Indeed, this is a much better option. A new version of Fig. 2 is embedded in the paper.

Reviewer: 2

General comments

The paper addresses a topical subject and has relevance. It seeks novel ways to determine how long deterrents remain effective.

Generally, I found this paper to make statements without really going into enough background literature at the international level to support the claims it does. For example, Page 2, line 1 “Despite of the huge effort in compiling evidence-based conservation interventions, those relevant for managing and protecting predators are still not incorporated [13]. This adds uncertainty and makes results of reviews of anti-predator interventions inconclusive [14].” This is very general to state that no literature incorporates managing and protecting predators are not incorporated is not true, and that reviews of anti-predator interventions are inconclusive is misleading. I suggest the latter is better broken down into why these may be inconsistent as there are a range of issues in this regard. This is important as the topic is complex dealing with many species across varying environmental conditions.

We have modified the text and added references to support these statements (also see the next paragraph).

I think that more studies are available to incorporate to this study to increase the value of this manuscript.

We are aware of many more publications about the effectiveness of anti-predator interventions and added a sentence in the beginning of Results about the numbers of papers found and used for this study. The issue is that other studies provided only one-off effectiveness data for the whole period of study, so it was impossible for us to track changes in the effectiveness over time. We had to use only the studies which contained data of predator-caused damage (e.g., livestock losses) during at least two periods of time so we could construct the trend lines with at least two data points (Fig. 2). If predators did not cause damage during the whole study, i.e. an intervention was 100% effective, we assumed that damage was nil also during the shorter periods of time and placed such cases in Table 2.

We added a sentence in Methods about the requirement of ≥ 2 data points to study inclusion. We also added a phrase in Discussion to explain that it is impossible to track temporal changes from one-off measurements of effectiveness.

Another aspect is that we took the standardized, most common metrics of damage (number of livestock individuals killed, number of beehive and crop damage records, and number of predator individuals resuming nuisance behaviour after an intervention) and did not use the studies which applied other metrics. For example, the paper by Davidson-Nelson and Gehring (2010) mentioned by the reviewer measured the effectiveness from the numbers of predator tracks in control and treatment (fladry) samples which is not equivalent to the numbers of livestock killed. Also, in this study livestock losses in control samples were zero and the relative risk of damage (RR) could not be calculated. However, we cited this work to describe the effectiveness of fladry.

The new methods are interesting and the authors should provide the code and models used to assist future work and make this comparative for future work.

No code is required for this approach as this is a rather simple statistical tool which can be applied in Excel. It is just required to collect data for the relative risk of damage (RR) and then calculate the % of damage reduction.

Overall, I found that the paper needs overall improvements in terms of 1) grammar throughout the manuscript, and 2) I found it a bit concerning that more studies that appear to suit the criteria to this review, were not included. Additionally, I think that the findings have value, however, the methods require more detail and providing the models will help readers understand this better.

We read our paper thoroughly and made corrections in the language when necessary. We also added more text to clarify details and support our results.

Abstract

It seems that the authors may not use English as a first language, and in its current form, the manuscript will require major grammatical editing throughout, by an editor with better experience with the language.

For example, page 2 lines 14-15, “To know for how long do interventions remain effective...” would better read “To know how long interventions remain effective for..”. Then again in lines 21 and 22, “We found electric fences...100% during up to three years” indicates little effort on editing this document. This continues throughout the manuscript and should be improved prior to publication.

We made corrections in this part and throughout the paper.

Page 1, line 22 – 23, “The effectiveness ...eroded quite fast after 1 – 5 months...” I would like to see the range or average here if that is possible.

The interval of 1-5 months provides a range of the breakpoint (threshold) time periods at which the effectiveness begins to decrease. As the trend lines of these interventions contained less than 10 data points, we could not apply the segmented regression analysis and estimate the mean and standard error of their breakpoints. Supplemental feeding and use of guarding llamas were the only interventions which provided large enough samples to conduct the segmented regression, as shown in the Results section.

Line 25 – 26, In which way was night corrals and herding inclusive? Was it inclusive of short term effectiveness? This is an odd finding compared to other literature. What would be the reasons for this in this case?

We could not determine how the effectiveness of night corrals and herding changed over time because we had only one case study from each of these techniques. However, we deleted this sentence from Abstract to keep it to the word limit.

Introduction

Line 33, I would suggest more international studies as references to this broad problem statement.

We added the most up-to-date and comprehensive references to the first sentences.

Line 39 – 41, “Human-predator conflict may occur...”. Firstly, the grammar is very poor. Also, these are certainly not the ONLY two instances carnivore-wildlife conflict occur. It is misleading in its current statement. Perhaps the authors are introducing some examples of when conflict occurs? This needs to be stated as examples, rather than the ONLY two scenario’s for human-carnivore conflict situations.

We agree that these two options are not the only causes of human-predator conflicts, but the meaning of this sentence was to relate conflicts to prey availability. We have changed the sentence to make the context clear.

Page 2, line one please see general comments section above.

We have modified the text and added references to support these statements.

Page 2, lines 7 – 10, suggests that there is no evidence in literature discussing duration of effect. There are more studies which assess and discuss depredation over time using various methods albeit for certain periods, but these have relevance to this manuscript. See below a few papers I found on the duration effect and I’m sure there will be more I didn’t have the time to look for more:

- Tools for the Edge: What's New for Conserving Carnivores

John A. Shivik *BioScience*, Volume 56, Issue 3, March 2006, Pages 253–259”. Looked at several tools with various findings of duration of effect.

- Non-lethal defence of livestock against predators: flashing lights deter puma attacks in Chile, Ohrens et al., 2019, found that fox lights worked for at least 4 months.

- Davidson-Nelson SJ and Gehring TM. 2010. Testing fladry as a nonlethal management tool for wolves and coyotes in Michigan. *Human-Wildlife Interactions* 4: 87–94.

- Gehring TM, VerCauteren KC, and Cellar AC. 2010. Good fences make good neighbors: implementation of electric fencing for establishing effective livestock protection dogs.

HumanWildlife Interactions 4: 144–49. Gehring TM, VerCauteren KC, Provost ML, et al. 2010. Utility of livestock-protection dogs for deterring wildlife from cattle farms. *Wildlife Res* 37: 715–21.

- Espuno N, Lequette B, Pouille ML, et al. 2004. Heterogeneous response to preventive sheep husbandry during wolf recolonization of the French Alps. *Wildlife Society B* 32: 1195–208.

The point above along with the couple examples I found relatively quickly, has relevance to page 2, lines 17 – 21.

We are well aware of these and other studies, but they estimated the effectiveness once for a fixed period of study duration. From the data available, it is not possible to break down the study period into shorter periods and to track how the predator-caused damage and the effectiveness of interventions changed over these shorter periods. Our requirement to study inclusion was to find studies which explicitly monitored the damage and effectiveness over several periods of time, at least two periods, so we could see how the effectiveness changes over time within the same study. We added a sentence in Methods about this and also clarified the text in Introduction.

The study by Ohrens et al. 2019 was published after our study was done.

The comparisons of different studies with different durations, e.g. study 1 with dog effectiveness over 6 months and study 2 with dog effectiveness over 12 months, are irrelevant as the conditions in study 1 and study 2 are by default different. We wanted to see the patterns of effectiveness changes just within each study.

We also excluded the studies which used different metrics of damage estimation, such as visitation rates from predator track counts (Davidson-Nelson and Gehring 2010). However, we cited this paper to describe the effectiveness of fladry. Further, we could not use the studies whose data were insufficient for calculations of RR, e.g. when damage records in control samples (B) were zero.

Lines 22 – page 3 lines 6 lead the reader more clearly into what is being assessed. I suggest the authors bring this in earlier in the introduction and summarise this better in the abstract, as I had difficulty in understanding the objectives until this point.

We modified the text in Introduction and Abstract to explain study objectives in a more straightforward way.

Methods

Page 3, line 18, why was the method limited to two search periods i.e. 2000-2005 and then from 2014-2016? Why was there a period gap and why did this search not look until 2018 like the other sites?

During this study, Carnivore Damage Prevention News was posted on www.lcie.org and contained the issues only from these two periods, 2000-2005 and 2014-2016. Later on, the collection was supplemented by the issues dated 2016-2018 and moved to www.medwolf.eu, but the gap of 2005-2014 still exists and most likely this newsletter was not produced during this period.

An important problem with ‘effectiveness’ studies is that many studies are not based on verified losses, rather opinion. It would be of value to compare these two groups of studies differently as many don’t consider opinion surveys realistic to the ‘effectiveness’ question due to bias or placebo effect for example.

We agree that this problem always exists in depredation studies and livestock losses assessed by owners can be true or provide underestimates or overestimates of true losses. Verification of owners’ estimates is very rarely possible for practical reasons (old records, researchers not available etc.) and this was a case also for our study. We did not separate verified and owner assessed data for several reasons. First, our sample size was small and we avoided to further split it apart. Second, separation of owner assessed data would not shed light on their accuracy and it would still remain obscure how reliable they are. Third, our primary goal was to find and explain trends and we believe that their representation in this study is reliable as the same method (verification or owner assessment) had been used to collect data for each study case and effectiveness trend line.

To estimate effectiveness, we did not use farmers’ subjective opinions like ‘very effective’, ‘effective’ or ‘not effective’ because they express perceived effectiveness, not true effectiveness. Instead, we always used the quantitative metrics of damage (No. livestock killed, No. damage

records and No. nuisance animals) that could be reliably compared in control vs. treatment samples. We added a sentence in Methods about this.

The algorithm designed for this is new and appears to be relatively transferable to other studies. However:

1) How was N_c data available in each of these cases? It is rarely listed in deterrent studies.

Ideally, effectiveness studies need a separate control sample and a treatment sample, but in deterrent studies this is usually not applied for practical reasons. The study by Beckmann et al. (2004) was the only one in our study to indicate the randomly assigned control and treatment samples. Deterrent studies have been usually designed as before-after studies showing how many predators from a fixed-size sample kill livestock/crop or exhibit nuisance behaviour before and after a deterrent is applied. So, in most of our deterrent studies $N_c = N_t$. In some cases (Gustavson et al. 1982; McManus et al. 2015), the ratios A/N_t and B/N_c were provided as the percentages and we did not know N_c as such. We added a sentence about this.

2) Building such models adds vagueness to the study results. I suggest that the models and code developed be added as a supplementary 2 document to assist future work. This may also assist researches to identify variables which cause more or less variation in results.

Our approach does not require models or codes, and the % of damage reduction can be easily calculated in Excel if the data of A, B, N_t and N_c are collected and stored in the dataset.

Discussion

Consider placing paragraph two lines 24 – 29 in results section.

In the pdf file of submitted manuscript, these lines describe the lower effectiveness of some electric fence applications due to methodological flaws. This is not a direct result of our study and we would like to retain it in Discussion.

Line 53, was it an expectation to find guarding animals more effective? Why?

We expected the effectiveness of guarding animals to be high enough because they have been used for millennia and local people widely believe that without guarding animals losses would be higher. We improved the sentence.

Line 55. Did you want to see 100% reduction in methods? That is a reach for any method, however, rather it may be useful for the reader to know if there was a significant decline, or a difference in decline in depredation over that period of time as opposed to total decline?

Yes, one of the main aims was to find interventions which cause 100% reduction in damage, and we placed them in Table 2. Another aim was to see whether the effectiveness trend lines fit the patterns displayed in Fig. 1. As all trend lines contained only one effectiveness estimate at a time and did not contain error margins, we could not estimate whether the changes in effectiveness were statistically significant or not.

A comparison of intervention-caused and general damage declines requires an application of the Before-After-Control-Intervention (BACI) approach. We found only one such case (Meadows

and Knowlton 2000) which described that a reduction of damage due to guarding llamas coincided in time with a decline of depredation losses in general, which might misleadingly conclude that llamas are ineffective in livestock protection. This sentence is in the beginning of Discussion.

Increasing the studies may help this manuscript. Studies do seem to be available relating to duration and livestock decrease over time in the list mentioned below are several, but more are available that are not included in this study. I understand it is not easy finding these, however, the small sample size does limit impact this paper can make. Also, it is important to find or separate studies which consider actual losses versus perceived losses.

Please see our responses above.